# Flubendiamide Resistance and Its Mode of Inheritance in Tomato Pinworm *Tuta absoluta* (Meyrick) (Lepidoptera: Gelechiidae)

**DOI:** 10.3390/insects13111023

**Published:** 2022-11-05

**Authors:** Lian-Sheng Zang, Zunnu Raen Akhtar, Asad Ali, Kaleem Tariq, Mateus R. Campos

**Affiliations:** 1Key Laboratory of Green Pesticide and Agricultural Bioengineering, Guizhou University, Guiyang 550025, China; 2Department of Entomology, University of Agriculture Faisalabad, Faisalabad 38000, Pakistan; 3Department of Entomology, Abdul Wali Khan University Mardan, Mardan 23200, Pakistan; 4INRAE, CNRS, UCA, 06903 Sophia-Antipolis, France

**Keywords:** *Tuta absoluta*, resistance inheritance, cross-resistance, autosomal, incomplete recessive, flubendiamide

## Abstract

**Simple Summary:**

Tomato pinworm, *Tuta absoluta* (Meyrick), is the major pest of tomato crops in Pakistan. To develop a better insecticide resistance management strategy and evaluate the risk of resistance evolution, a field collected population of tomato pinworm was selected with flubendiamide in the laboratory. We investigated the genetics of flubendiamide resistance by selecting a field strain of tomato pinworm with commercial flubendiamide formulation and dose-response mortality of flubendiamide-selected generation to other insecticides. The flubendiamide-selected (Fluben-sel) strain demonstrated a higher concentration-mortality response against chlorantraniliprole, thiamethoxam, permethrin, abamectin and tebufenozide compared to the unselected population. The backcross analysis of F_1_× resistant parent suggests that resistance is controlled by more than one factor. Resistance progression from 38 to 520 folds demonstrated that *T. absoluta* can develop a higher level of resistance. These results could be helpful to design resistance management strategies for the tomato pinworm.

**Abstract:**

Tomato pinworm, *Tuta absoluta* (Meyrick) (Lepidoptera: Gelechiidae) is the major pest of tomato crops in Pakistan. Insecticides are commonly used for the management of this insect-pest. To develop a better insecticide resistance management strategy and evaluate the risk of resistance evolution, a field collected population of the tomato pinworm was selected with flubendiamide in the laboratory. We investigated the genetics of flubendiamide resistance and concentration-mortality response to other insecticides by selecting a field strain of tomato pinworm with commercial flubendiamide formulation. *Tuta absoluta* was reciprocally crossed with resistant strain (Fluben-R) and was selected up to 13 generations, while F_1_ progeny was back-crossed with resistant parent (Fluben-R). The results of LC_50_ and Resistance Ratio (RR) demonstrated a higher resistance developed in field and laboratory-selected strains (G_2_ and G_13_, respectively). Field-collected and laboratory-selected (Fluben-R) strains demonstrated higher intensity of concentration-mortality response against chlorantraniliprole, thiamethoxam, permethrin, abamectin and tebufenozide compared to susceptible ones. Based on the overlapping of 95% FL, it demonstrated significant differences, revealing that it was not sex linked (autosomal) with no maternal effects. The backcross analysis of the F_1_× resistant parent resulting in significant differences at all concentrations suggests that resistance is controlled by more than one factor; the null hypothesis was rejected and inheritance was under polygenic control. Resistance progression from 38 to 550 folds demonstrated that *T. absoluta* can develop a higher level of resistance to flubendiamide. Concentration-mortality response experiments demonstrated that the LC_50_ of some tested insecticides was higher for field-collected and laboratory-selected strains, suggesting that resistance mechanisms should be studied at a molecular level for better understanding. These results could be helpful to design resistance management strategies against the tomato pinworm.

## 1. Introduction

Tomato pinworm, *Tuta absoluta* (Meyrick) (Lepidoptera: Gelechiidae), is endemic to tropical and subtropical regions of the Americas [1,2,3]. From its Peruvian origin [4], this species remained restricted in South America until the mid-2000s [5]. In addition to the difficulty of its control and high invasive capacity characteristic of its phenotypic plasticity [6], it has turned into a truly global pest after expanding its geographic range to the Europe [3,7] and Africa regions [3,8,9,10,11,12,13,14].

Several models have been predicting the high capability of tomato pinworm to invade new areas [14,15,16]. By 2017, the tomato pinworm had spread from the western Palearctic to the Himalayan [14,17,18,19,20], and by 2021, it had already reached Western China [21]. This alarming increase in pinworm infestation poses threats to domestic production as well as international exports of fresh tomatoes [22,23]. Pakistan usually imports tomato during the shortage season, which always leads to the spread of tomato pests. In Pakistan, the first report of tomato pinworms was published in 2016 [24], followed by a second report in 2020 [25] demonstrating the concern over the spread of the pest to Solanaceous crops, such as eggplant, potato, and tobacco [25,26]. The tomato pinworm exhibits cryptic behavior that seriously compromises tomato yield depending on the year, season, and pest density [27], making pest management efforts quite challenging to control this pest [28]. Reviewed by Biondi and Desneux [29], the invasive pest attacks cost accounts for around 70 billion USD per year [30]. Sincelower tomato yields are the results of tomato pinworm damage, which includes leaf mining and fruit infestation [1,3]. Therefore, if no control measures are taken, tomato pinworm can cause up to 70–80% yield loss in tomato crops and will continue to remain a threat to greenhouse and open-field tomato production [1].

Currently, the primary method of controlling tomato pinworm is the quarantine measures that help in minimizing its entry into new areas. However, once the border inspections have been crossed, and the pest established itself in the new area, chemical control becomes the principal tool of controlling tomato pinworms [31,32]. However, an additional phytosanitary concern is that the introduced pest’s phenotype could include inheritable traits that could impose management difficulties, including insecticides’ resistance [6]. Thus, the possible presence of resistance alleles and the lack of effective insecticides due to the lack of records at the place of introduction can facilitate the rapid spread of tomato pinworm.

Following the failure of the initial efforts to control the tomato pinworm through phytosanitary methods, insecticides end up being the most popular strategy utilized to contain the tomato pinworm in new areas. There is a tendency for the average number of insecticide treatments and expenditures related to pest management to increase significantly due to a lack of preliminary investigations on registered pest-specific insecticides [33]. Consequently, the frequent use of insecticides will invariably lead to the failure to offer adequate control levels, emphasizing the need for novel insecticides that are specifically targeted against tomato pinworm. However, new insecticides can become ineffective if there are no other active ingredients to alternate with them in pest management, which may promote pest resistance to insecticides [34,35,36,37,38,39,40]. Furthermore, even though the studies are rigorous, they cannot fully cover all potential side effects on non-target organisms [41] and human health [42].

Diamides have been overused since their launch to the market because tomato pinworm populations were initially highly susceptible to them [43]. Consequently, a number of control failures have quickly emerged in the field with high resistance recently reported [3,44,45,46]. Diamides were launched to the market with initial registrations for the management of *Plutella xylostella* in Southeast Asia [47]. The mode of action of diamides is that it binds to the insect’s ryanodine receptors; the calcium channels mediating calcium release from intracellular stores in neuromuscular tissue result in muscle contraction [48]. In Asia, Brazil, and EUA, *P. xylostella* was the first insect to develop resistance to diamide insecticides [49,50]. Currently, diamide resistance was already detected in the diverse population of complex *Spodoptera*, *Adoxophyes honmai,* and *Chilo suppressalis* [51,52,53,54,55,56,57,58,59]. Surveys after found the first occurrences of diamide resistance in greenhouse and field populations of tomato pinworm in Brazil [50], Kuwait [60], and Italy [61], also confirming prior reports of altered target-site resistance [62,63,64].

There is a discrepancy between studies on the genetic inheritance of tomato pinworm to chlorantraniliprole. Tomato pinworm showed the resistance inheritance autosomal, completely recessive and monofactorial in Brazil. Whereas, in the study by Roditakis et al., the tomato pinworm demonstrated a resistance inheritance incompletely recessive and polygenic [63]. However, both studies agreed that the selection of population was chosen because of its high level of resistance to chlorantraniliprole. In addition, it demonstrated a cross-resistance to other diamides such as cyantraniliprole and flubendiamide. Moreover, the investigation of genetic inheritance resistance to flubendiamide can contribute to accurately elucidate the characteristics of tomato pinworm resistance to diamides. Therefore, the objective of this study was to (i) elucidate the pattern of inheritance of flubendiamide resistance in the tomato pinworm; and (ii) determine the cross-resistance spectrum, if encompassing diamides and even other classes of insecticides. This study could be helpful to design suitable resistance management strategies for the control of tomato pinworm.

## 2. Materials and Methods

### 2.1. Insect Collection

Both insect groups, the resistant (collected from the flubendiamide-sprayed tomato field) and susceptible strains (collected from the unsprayed field over years and no tomato crop was sown in the vicinity) were collected during 2018, 2019, and 2020 from four different districts (Lahore, Faisalabad, Multan, and Sahiwal) of Punjab, Pakistan, to know the field scenario of the tomato pinworm resistance and susceptibility. Laboratory conditions were maintained at 27 ± 2 °C of temperature, with 65% RH, and a 14:10 h L:D photoperiod.

For inheritance and cross resistance studies, four thousand (n = 4000) larvae were collected from tomato fields sprayed with flubendiamide in the Faisalabad district, were brought to the laboratory and were kept in separate cages of growth chambers with a temperature of 27 ± 2 °C, 65% relative humidity, and 14:10 h L:D photoperiod. The field population was kept until the second generation (G_2_) on tomato sprayed with flubendiamide. To obtain the selected population under the laboratory, the field-collected population was selected with flubendiamide for 13 generations, while the designated susceptible strain (collected from an un-sprayed tomato field over years) was kept on non-sprayed tomato for a year.

### 2.2. Insecticides Formulations and Flubendiamide Selection of Insects

Commercially available insecticides including flubendiamide (Belt 480 SC), chlorantraniliprole (Coragen 20 SC), thiamethoxam (Actara 25% WG), permethrin (Adpex 25% EC), abamectin (Proclaim 19 EC), and tebufenozide (Topgun 200 g a.i./L SC) were used for bioassays. Resistant strains of third instar larvae were reared on flubendiamide sprayed tomatoes for 13th generations. From the 1st generation, the selection was performed at different concentrations and a large number of larvae were maintained to obtain further generations of selection. Each generation was reared on a subsequent increased concentration of flubendiamide. The designated susceptible strain was kept on tomato without spray for one year. For a concentration-mortality response to a different insecticide experiment, larvae were taken from resistant (G_13_) and susceptible colonies.

### 2.3. Bioassay of Larvae

Tomatoes were used for resistance and cross-resistance experiments. In these experiments, six serial dilutions with three replications for each were used. For colonies, insecticide sprayed on diet with concentration of 0 to 15 µg a.i./mL for susceptible and 45 to 400 µg a.i./mL was sprayed on a diet of resistant insects during G_1_ to G_13_. For one replication thirty (n = 30) larvae were used and fresh tomatoes, dipped in insecticides and air dried for 10 min, were replaced every day.

### 2.4. Genetic Reciprocal and Back Crosses

The resistant strain was collected from a sprayed tomato field, while the susceptible strain maintained in the laboratory without insecticide exposure was collected from tomato fields, where no spray was used over the years. To determine resistance patterns against flubendiamide in *T. absoluta*, susceptible and resistant strains were assumed to be homogenously susceptible and resistant. Reciprocal crosses were conducted by non-mated males for 6 h in plastic jars. Non-mated adults were immediately separated on a basis of sex and were added in other jars as pair. In this way, two types of reciprocal crosses of Resistant 
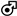
 × Susceptible 
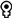
 and Resistant 
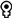
 × Susceptible 
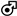
 were conducted. F_1_ progeny was further back crossed with a resistant parent in a reciprocal way. Each time 300 pairs were taken and were kept in separate cages for further bioassays.

### 2.5. Maternal Effects and Sex Linkage

Maternal effects and sex linkages of resistance from slope and fiducial limit (FL) of LC_50_ for F_1_ progeny (which was already obtained from the reciprocal crosses of resistant and susceptible strains) were determined. LC_50_ values were considered significantly different if there was no overlap of 95% FL.

### 2.6. Degree of Dominance

The dominance values were determined using formula [65,66].
D = (2XF − XRR − XSS)/(XRR − XSS)

XF, XRR, XSS are LC_50_values of F_1_, Fluben-Sel and Susceptible populations, respectively, of reciprocal progeny.

The degree of dominance values ranged from −1 (completely recessive) to 1 (completely dominant resistance) [65,67,68].

### 2.7. Number of Loci Influencing the Inheritance

Reciprocal crosses of F_1_ and back-crossed progeny was conducted to determine the number of loci influencing the inheritance. To test the null hypothesis, if resistance was controlled by one locus with two alleles; Resistant I and susceptible (S), then the parental R is 100% RR and F progeny is 100% RS. The back cross of RS × RR will produce 50% RS and 50% RR. The concentration of x for RR × RS backcross can be calculated as of [69]:Y_X_ = (M_RS_ + M_RR_)/2

While M_RS_ and M_RR_ are mortalities of the presumed RS and RR genotypes at concentration x, the chi-squared values were calculated as follows [70].
χ^2^ = (F_1_ − pn)^2^/pqn

F_1_ = observed mortality of backcross at concentration x, n = number of total progenies exposed to concentration, p = expected mortality, and q = 1 *−* p. Then, the sum of χ^2^ is at each concentration compared with the chi-squared with one degree of freedom. The inheritance of resistance will be considered as fit for the monofactorial model if df = 1.

### 2.8. Statistical Analysis

POLO program [71] was used for probit analysis [72] based on the concentration response from each Mendalian cross. LC_50_ demonstrated 50% larval mortality with a 95% Fiducial Limit (FL). LC_50_ for reciprocal crosses was considered significant when the Fiducial Limit (FL) did not overlap [73]. Probability calculation was based on (χ^2^) Chi-square. The Resistance Ratio (RR) with 95% FL was calculated using the method of Robertson and Preisler, and considered significant if it did not include 1 [74].

## 3. Results

### 3.1. Field Strains’ Resistance Evaluation

In Table 1, it is evident that 3 consecutive years (2018, 2019, and 2020) of the data of flubendiamide resistant field-collected strains demonstrated a higher resistance ratio (RR) in four districts (Lahore; Faisalabad; Multan; Shahiwal) of Punjab, Pakistan (Table 1). For the Lahore district-collected population, LC_50_ was 49.57, 51.76, and 50.56 µg a.i./mL and RR was 51.10, 43.13, 44.35 folds during 2018, 2019, and 2020, respectively. For the Faisalabad district-collected population, LC_50_ was 52.35, 51.49, and 50.69 µg a.i./mL and RR was 44.74, 55.96, 44.07 folds during 2018, 2019, and 2020, respectively. The Multan district-collected population LC_50_ was 54.38, 55.09, and 52.67 µg a.i./mL and RR was 46.87, 59.88, and 49.68 folds during 2018, 2019, and 2020, respectively (Table 1). The Sahiwal district-collected population LC_50_ was 56.48, 50.58, and 54.72 µg a.i./mL and was 45.54, 46.40, and 55.83 folds in the 3 consecutive years (Table 1).

### 3.2. Laboratory Selection of Resistance to Flubendiamide

The tomato pinworm population was selected until 13 generations and mortality decreased gradually with an increase in generations, as is inversely proportional. The concentrations’ range of flubendiamide was 80 to 1000 µg a.i./mL, resulting in mortality from percentages from 66.6% to 0.19% (Table 2). The selection with flubendiamide until 13 generations resulted in an increase in LC_50_ ranging from 49.50 to 709.81 µg a.i./mL (Table 3). Meanwhile, the RR increased from 38.37 to 520.24 folds as compared to the susceptible (Table 3).

### 3.3. Maternal Effects

From Table 4 and Table 5, LC_50_ of F_1_ progeny having reciprocal cross of susceptible and resistant strains (50.32 and 51.74) showed an overlapping of 95% FL, which exhibit that there was non-significant difference (Table 4), proving that it was not sex linked (autosomal) and having no maternal effects. Further LC_50_ of back-crosses with resistant parent showed 68.63, 65.40, 63.67, and 61.91 (µg/mL). Resistance Ratio (RR) values for the back-cross were 46, 43, 42, and 40 folds (Table 5).

### 3.4. Loci Influencing Inheritance for Monogenic Fit Test

Actual and expected mortality for determining the mode of resistance involved in the tomato pinworm using the chi-square fit [70] for the Mendalian single gene model of resistance was conducted. The results demonstrated that significant differences at all concentrations occurred and actual mortality was higher than the expected mortality for all concentrations in all crosses (*p* < 0.0001), demonstrating that multiple factors controlling the resistance to flubendiamide in tomato pinworm; the null hypothesis was rejected and the inheritance was under polygenic control (Table 6). 

### 3.5. Degree of Dominance

Dominance results demonstrated that the percent survival was decreased when flubendiamide concentration was increased (0.5–70 µg/mL). The Dvalue demonstrates that resistance was expressed as a complete dominant at 1 (Table 7).

### 3.6. Concentration Mortality Response of Different Insecticides

Laboratory-selected strains LC_50_ values were 132.67, 188.46, 171.69, 175.89, and 152.30 µg a.i./mL for chlorantraniliprole; thiamethoxam; permethrin; abamectin; tebufenozide, respectively (Table 8). The laboratory-selected resistant strains demonstrated the highest decrease in susceptibility to thiamethoxam. While in cases of abamectin; permethrin; tebufenozide; chlorantraniliprole, it demonstrated a decreasing trend in susceptibility (Table 8).

## 4. Discussion

Effective management strategies for controlling insects that are detrimental to field crops can be developed by understanding the resistance development and mode of inheritance. Our investigation on the efficacy of flubendiamide against the tomato pinworm indicated that *T. absoluta* exhibited high levels of resistance to diamides as well as dose response mortality to several chemical insecticide groups. The D value dominance of resistance reduced from 1 to 0.31 with a higher concentration. According to our findings, the mode shifts from complete dominant to incomplete recessive as the concentration increases. However, the tomato pinworm can develop a high level of resistance when receiving continuous treatment in the laboratory, as evidenced by the *h* value of 0.31 at the highest concentration in the13th selected generations of flubendiamide. Furthermore, due to different selection histories, different genetic basis occur in insect populations. According to a prior study, the increase in insecticide concentration can alter the degree of dominance [75]. These findings suggest that with the increased concentration of flubendiamide, resistant alleles may increase in frequency under laboratory conditions. Although the level of dominance for a particular character is a fixed parameter, environmental conditions or genetic background may also influence the dominance level [76,77]. Sometimes dominance level may undergo evolution because of the selection of insecticide, and selection may favor resistant alleles conferring more dominant phenotypes through allele replacement [78]. Similarly, the rate of insecticide resistance development could also be affected by the degree of dominance; resistance evolves more rapidly when conferred by dominant or incompletely dominant genes than when regulated by recessive genes. Resistant genotypes including heterozygotes have higher survival ability after pesticide application in the field and can multiply faster than susceptible.

Flubendiamide resistant field population of tomato pinworm has demonstrated dose-response mortality to chlorantraniliprole, thiamethoxam, permethrin, abamectin, and tebufenozide. Flubendiamide is the first diamide that belongs to the phthalic acid derivatives and has same mode of action as chlorantraniliprole [79,80,81]. Previous studies with *Plutella xylostella* and *Adoxophyes honmai* demonstrate resistance to flubendiamide [51] and chlorantraniliprole [52], respectively. In addition, high cross-resistance between flubendiamide and chlorantraniliprole has also been reported under field conditions [61,82]. In contrast to a previous study, which found limited susceptibility to abamectin and permethrin against tomato pinworm [83], our concentration-mortality response data demonstrated increased LC_50_ for abamectin and permethrin.

Long-term sustainability of a single method of controlling tomato pinworms is impossible. Therefore, it is highly advised to adopt the integrated techniques of control to maximize results while avoiding environmental problems [3,84]. In addition, a long-term strategy should be developed combining biological control, plant resistant, and selective insecticides [1,7]. This notion was reinforced with current study findings that the aforementioned pesticides should not be sprayed after flubendiamide in tomato fields.

The reciprocal crosses of susceptible and resistant strains and the back cross of F1 progeny with a resistant parent, resulting in the 2nd generation, demonstrated no significant difference, suggesting an autosomal inheritance to flubendiamide [85]. Previous research on the genetic basis of tomato pinworm resistance to spinosyns indicated autosomal resistance to spinosad with cross-resistance to spinetoram. [6]. These reciprocal and backcrosses in our experiments demonstrated a many folds resistance ratio [63,64], suggesting that marginal effects may occur due to the bioassay rather than maternal sex linkage. The results of inheritance demonstrated that resistance was incomplete recessive, not sex linked and autosomal in nature. Furthermore, resistance could be controlled by several factors [63,64], as demonstrated by the discovery that chlorantraniliprole exhibits a resistance influenced by multiple traits.

Insect populations may confer monogenic or polygenic resistance to insecticides under high selection pressure, and polygenic resistance is more likely to happen in this situation [86]. Resistance to flubendiamide was polygenic-controlled by more than one factor. Based on the monofactorial model, when the number of genes gives greater value than 1, it means multiple factors are involved in the flubendiamide resistance in the tomato pinworm. A higher chi-square value often demonstrates that the monogenic model is rejected, and resistance is controlled by more than one gene. Field evolved laboratory selected populations revealed polygenic behavior due to variations in natural selection. Meanwhile, monogenic resistance occurs in natural populations of the field in the same vicinity [76,87,88]. Polygenic resistance may occur due to the difference in selection history with a separate mechanism of resistance, as of the insect species [89]. Our results of polygenic resistance with the dominance mode are in accordance with studies on *Plutella xylostella* demonstrating a similar trend against abamectin [90].

In conclusion, the resistance of flubendiamide against *T, absoluta* found an autosomal, is completely dominant and is controlled by more than one factor, suggesting that *T, absoluta* has a considerable potential to develop a higher level of resistance against flubendiamide. Dose-response mortality against chlorantraniliprole, thiamethoxam, permethrin, abamectin and tebufenozide suggests that these insecticides should be avoided in rotation with flubendiamide in pest management programs, and this strategy could be helpful to reduce the selection pressure and resistance development in *T, absoluta*.

## Figures and Tables

**Table 1 insects-13-01023-t001:** Resistance levels of *Tuta absoluta* field populations to flubendiamide.

Field Area	Strains	Year	n ^a^	LC_50_ (95%FL) (µg/mL)	Fit for Probit Line		RR ^a^
					Slope ± SE	χ^2^ (df = 8)	
Lahore	Fluben-S	2018	810	0.97 (0.69–1.33)	1.67 ± 0.32	1.26	1
		2019	780	1.20 (0.89–1.57)	2.37 ± 0.46	1.41	1
		2020	832	1.14 (0.79–1.48)	1.43 ± 0.52	1.23	1
	Fluben-R	2018	835	49.57 (23.51–69.84)	2.37 ± 0.82	2.29	51.10
		2019	704	51.76 (34.65–80.47)	1.86 ± 0.62	2.30	43.13
		2020	785	50.56 (32.31–74.69)	2.62 ± 0.89	1.76	44.35
Faisalabad	Fluben-S	2018	843	1.17(0.89–1.62)	2.59 ± 1.14	2.14	1
		2019	827	0.92 (0.59–1.46)	1.89 ± 0.59	1.57	1
		2020	794	1.15 (0.79–1.69)	1.62 ± 0.72	2.54	1
	Fluben-R	2018	806	52.35 (33.46–71.23)	2.60 ± 0.41	2.32	44.74
		2019	813	51.49 (38.47–71.42)	2.34 ± 1.03	2.50	55.96
		2020	789	50.69 (31.37–76.64)	1.54 ± 0.72	1.07	44.07
Multan	Fluben-S	2018	816	1.16 (0.79–1.54)	2.33 ± 0.65	0.97	1
		2019	819	0.92 (0.57–1.49)	1.47 ± 0.53	1.75	1
		2020	779	1.06 (0.72–1.70)	2.39 ± 0.81	1.41	1
	Fluben-R	2018	819	54.38 (31.26–91.34)	2.86 ± 0.93	2.92	46.87
		2019	788	55.09 (29.65–88.45)	1.29 ± 0.69	1.89	59.88
		2020	830	52.67 (31.04–89.75)	2.37 ± 1.09	1.73	49.68
Sahiwal	Fluben-S	2018	817	1.24 (0.69–1.67)	2.10 ± 0.80	2.56	1
		2019	790	1.09 (0.74–1.51)	1.35 ± 0.82	1.74	1
		2020	809	0.98 (0.48–1.52)	2.12 ± 0.21	2.58	1
	Fluben-R	2018	832	56.48 (31.37–98.60)	2.63 ± 1.20	1.53	45.54
		2019	821	50.58 (27.24–93.46)	1.28 ± 0.76	2.49	46.40
		2020	798	54.72 (29.45–87.65)	2.37 ± 1.14	2.52	55.83

**Table 2 insects-13-01023-t002:** History of generations selected with flubendiamide and percent mortalities of *Tuta absoluta*.

Generation	Concentration (µg/mL)	No. of Larvae Exposed	No. of Larvae Dead	Mortality (%)
G1	80	1800	1200	66.60
G2	120	1000	100	10.00
G3	170	1100	30	2.72
G4	250	1050	28	2.54
G5	300	950	20	2.10
G6	450	1050	14	1.33
G7	500	920	19	2.06
G8	600	1100	11	1.00
G9	700	950	6	0.63
G10	800	1100	5	0.45
G11	900	1000	3	0.30
G12	1000	950	2	0.21
G13	1000	1050	2	0.19

**Table 3 insects-13-01023-t003:** The resistance levels of *Tuta absoluta* to flubendiamide during the selection process.

Selection	LC_50_ (95% FL) (µg/mL)	Slope ± SE	χ^2^	df	RR
Susceptible	1.29 (0.88–1.79)	2.81 ± 0.34	19.45	8	1
Flubendiamide-sel (G1)	49.50 (32.03–63.52)	4.30 ± 0.28	16.62	8	38.37
Flubendiamide-sel (G2)	67.32 (48.28–81.05)	3.69 ± 0.53	18.72	8	52.18
Flubendiamide-sel (G3)	94.35 (56.46–171.24)	3.52 ± 0.35	13.52	8	73.13
Flubendiamide-sel (G4)	157.51 (123.61–195.74)	2.60 ± 0.52	14.19	8	122.10
Flubendiamide-sel (G5)	213.73 (186.36–262.52)	3.45 ± 0.46	9.52	8	165.68
Flubendiamide-sel (G6)	302.67 (282.46–353.62)	4.68 ± 0.36	39.46	8	234.62
Flubendiamide-sel (G7)	389.54 (348.55–434.56)	2.89 ± 0.42	33.57	8	301.96
Flubendiamide-sel (G8)	446.61 (409.24–476.34)	3.59 ± 0.48	28.58	8	346.20
Flubendiamide-sel (G9)	496.26 (451.91–534.64)	3.73 ± 0.28	8.75	8	384.69
Flubendiamide-sel (G10)	562.72 (539.63–598.74)	2.84 ± 0.47	7.46	8	436.21
Flubendiamide-sel (G11)	613.44 (588.36–648.76)	3.57 ± 0.53	9.45	8	475.53
Flubendiamide-sel (G12)	658.77 (632.42–691.46)	3.64 ± 0.56	12.70	8	510.67
Flubendiamide-sel (G13)	709.81 (684.54–742.83)	4.75 ± 0.61	8.79	8	520.24

RR = LC_50_ of Resistant selected strain/susceptible strain.

**Table 4 insects-13-01023-t004:** Maternal sex linkage to determine the heredity involvement in *Tuta absoluta*.

Strain	LC_50_ (95% FL) (µg/mL)	Slope ± SE	χ^2^ (df = 8)
Susceptible	1.43 (0.81–1.97)	4.31 ± 0.73	12.31
Flubendiamide-sel (G13)	709.81 (684.54–742.83)	4.75 ± 0.61	8.79
Flubendiamide-sel 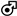 × S 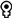	50.32 (42.35–61.32)	3.84 ± 0.27	10.43
S 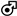 × Flubendiamide-sel 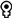	51.74 (40.51–61.86)	4.52 ± 0.59	12.60

Resistance will be considered significantly different if LC_50_ will not overlap on 95% Fiducial Limit and non-significant if LC_50_ will overlap on 95% Fiducial Limit. 
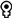
 and 
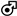
 represent female and male *T. absoluta*, respectively. The same as followed.

**Table 5 insects-13-01023-t005:** F_1_ progeny backcross with resistant parents of *Tuta absoluta*.

Strain	LC_50_(95% FL) (µg/mL)	Slope	χ^2^	RR
Susceptible	1.53 (0.96–2.07)	4.53 ± 0.26	11.32	1
F_1_ 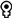 (S 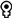 × Flubendiamide-sel 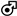 ) × RR2642	68.63 (47.29–83.13)	3.96 ± 0.44	9.27	46
F_1_ 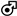 (S 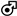 × Flubendiamide-sel 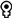 ) × RR 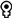	65.40 (46.27–79.20)	4.45 ± 0.53	8.83	43
RR 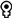 × F_1_ 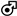 (S 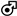 × Flubendiamide-sel 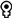 ) 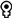 )	63.67 (49.88–75.80)	3.49 ± 0.34	11.52	42
RR 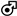 × F_1_ 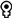 (S 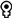 × Flubendiamide-sel 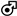 ) 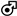 )	61.91 (42.36–74.69)	3.32 ± 0.72	8.96	40

**Table 6 insects-13-01023-t006:** Monogenic model for actual and expected mortality of *Tuta absoluta*.

Strain	Actual Mortality (%)	Expected Mortality (%)	χ^2^
F_1_ 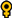 (S 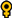 × Flubendiamide-sel 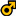 ) × RR 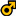			
20	8.73	0.5	1
40	28.56	12.31	0.49
80	74.32	17.69	0.54
100	100	35.43	0.26
F_1_ 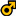 (S 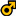 × Flubendiamide-sel 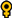 ) × RR 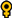			
20	7.94	0.5	1
40	26.24	10.58	0.40
80	73.04	19.08	0.64
100	100	30.25	0.35
RR 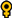 × F_1_ 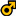 (S 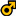 × Flubendiamide-sel 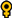 )			
20	7.13	1.54	0.063
40	27.03	11.32	0.21
80	75.50	18.41	0.34
100	100	30.31	0.52
RR 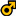 × F_1_ 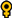 (S 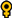 × Flubendiamide-sel 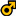 )			
20	8.62	1.61	28.32
40	27.93	11.91	0.24
80	76.37	24.72	0.62
100	100	31.40	0.39

Probability values were considered significantly different at *p* < 0.05.

**Table 7 insects-13-01023-t007:** Dose dependent effective dominance of flubendiamide-sel population of *Tuta absoluta*.

Concentration	Strain	Survival %	Fitness	D
	Susceptible	53.08	0.4	
0.5	Flubendiamide-sel	100	1	1
	F1	100	1	Complete dominant
	Susceptible	2.17	0.72	
2.5	Flubendiamide-sel	100	1	0.72
	F1	100	1	In-complete dominant
	Susceptible	0	0	
35	Flubendiamide-sel	51.32	1	0.62
	F1	21.23	0.62	Co-dominant
	Susceptible	0	0	
70	Flubendiamide-sel	13.41	1	0.31
	F1	0.41	0.31	Incomplete recessive

Effective dominance will be completely recessive at 0.

**Table 8 insects-13-01023-t008:** Concentration mortality response of flubendiamide-sel (G_13_) *Tuta absoluta* population to different insecticides.

Strain	Insecticide	LC_50_ (µg/mL)	Slope	χ^2^	Df
Susceptible	Flubendiamide	1.19 (0.74–1.89)	1.81 ± 0.71	9.41	8
Flubendiamide-sel (G_13_)	Chlorantraniliprole	132.67 (85.58–213.69)	3.47 ± 0.78	23.56	8
	Thiamethoxam	188.46 (125.76–246.79)	3.94 ± 1.46	21.46	8
	Permethrin	171.69 (134.57–260.48)	2.70 ± 0.63	17.31	8
	Abamectin	175.89 (123.81–278.70)	2.51 ± 1.74	21.69	8
	Tebufenozide	152.30 (92.60–215.72)	3.47 ± 1.26	16.75	8

## Data Availability

The data presented in this study are available on request from the corresponding author.

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
