# Peer review of "Flubendiamide Resistance and Its Mode of Inheritance in Tomato Pinworm Tuta absoluta (Meyrick) (Lepidoptera: Gelechiidae)"

_insects, 2022, doi:10.3390/insects13111023_

Round 1
Reviewer 1 Report
In this study, a flubendiamide resistant strain of Tuta absoluta (Meyrick) was selected from a field strain, and the cross-resistance pattern and inheritance mode were investigated. This is a valuable and meaningful study, but some research methods require more detailed statements and provide reliable references. According to the results of Table 3, if the flubendiamide resistant strain (Flubendiamide-sel (G13)) was selected from the Field-pop., the resistance ratio was just 14.88-fold compared with the reference strain. Moreover, further cross-resistance and inheritance research all should be based on these two strains which have the same genetic background. I do not think that this current manuscript is suitable for publication. I have the following comments.
1. According to the title, the realized heritability of flubendiamide resistance in the Tuta absoluta has been studied, however any results of realized heritability can't be found in the manuscript. In contrast, the inheritance mode of flubendiamide resistance in the Tuta absoluta was investigated.
2. In the part of 2.3 (line 131-141 of page 3), whether you choose a field collected resistant strain as an original population for flubendiamide resistance selection, or mix all the collected resistant strains for resistance selection? Please make it clear in the manuscript.
3. In the table 3, the susceptible, field-pop., unselected and Flubendiamide-sel strain should be clearly described to the reader, because the relationship between these strains are unclear.
4. Cross-resistance refers to the resistance of insects to one particular insecticide that may cause resistance to other insecticides that they have never been exposed to before. So, a selected resistant strain and an unselected reference strain which has same genetic background should be choose for cross-resistance study.
5. In the part of “2.5 Genetic reciprocal and back crosses”, genetic reciprocal and back crosses, reciprocal crosses and backcross lines should be established by resistance and susceptible strains with the same genetic background. Is your resistant population here screening for the susceptible strain?
6. In the part of “2.7. Degree of dominance Degree of dominance”, how do you calculate the fitness? Please provide detailed methods and references to support the evaluation of the fitness and degree of dominance.
7. In the part of 2.8 (line 167-180 of page 4), the references should be cited to support these two formulas.
8. In the part of “2.9. Statistical Analysis”, Fiducial limit (FL) and Confidence Interval (CI) are two different forms of expression, please keep one form in the manuscript.
9. Revise the title of table 1 to "Resistance levels of Tuta absoluta field populations to flubendiamide".
10. In the table 1, revise to "χ²(df)" and merge the two columns of data. When calculate the insecticide resistance ratio of field pest populations, a relatively fixed baseline of sensitivity needs to be established or cited from other study. In your this study, perhaps you can designate a relatively sensitive field population as a sensitive population and establish a sensitive baseline.
11. In the table 3, revise LC50 to LC50 (95%FL), revise Slope to Slope ± SE, two RR in the table, maybe you can change one RR to RR'.
12. In the part of “3.3 Maternal effects”, the results of Table 6 have not been presented and explained in the text, so, what can we get from Table 6? Moreover, the backcross information is not completed in the table 6, please revise it.
13. In the table 5, the LC50 values of parental and reciprocal crosses (F1) should be listed here, but the value of resistant strain is absent. Please revise the LC50 to LC50(95%FL), revise Slope to Slope ± SE, and merge the two columns of χ² and df to χ² (df).
14. In the part of "3.4. Loci influencing inheritance for monogenic fit test", according to the results of Table 7, I can't get any information of statistical analysis.
15. In the part of "3.5. Degree of dominance", I can't understand these findings, does the degree of dominance here is represent the flubendiamide resistance in the Tuta absoluta? If so, how does the degrees of dominance of the resistance can be changed under different concentration of flubendiamide exposure.
16. In the part of "3.6. Cross-resistance", the tables in the manuscript should be numbered in an order. The cross-resistance ratio (CR = LC50 of flubendiamide-resistant strain/LC50 of the unselected strain) has usually been used for evaluating the cross-resistance level between different pesticides.
Author Response
In this study, a flubendiamide resistant strain of Tuta absoluta (Meyrick) was selected from a field strain, and the cross-resistance pattern and inheritance mode were investigated. This is a valuable and meaningful study, but some research methods require more detailed statements and provide reliable references. According to the results of Table 3, if the flubendiamide resistant strain (Flubendiamide-sel (G13)) was selected from the Field-pop., the resistance ratio was just 14.88-fold compared with the reference strain. Moreover, further cross-resistance and inheritance research all should be based on these two strains which have the same genetic background. I do not think that this current manuscript is suitable for publication. I have the following comments.
Answer: Ok, we have added more clear methods of selection of resistant field population and its selection in laboratory, more references have also been added in Material and Method section under Insect collection sub-heading.
- According to the title, the realized heritability of flubendiamide resistance in the Tuta absoluta has been studied, however any results of realized heritability can't be found in the manuscript. In contrast, the inheritance mode of flubendiamide resistance in the Tuta absoluta was investigated.
Answer: Ok, we have deleted “Realized heritability” from title and “inheritance” has been added in title. We agree with comments that this work is inheritance mode of resistance of fludendiamide.
- In the part of 2.3 (line 131-141 of page 3), whether you choose a field collected resistant strain as an original population for flubendiamide resistance selection, or mix all the collected resistant strains for resistance selection? Please make it clear in the manuscript.
Answer: Field collected population was selected in laboratory for 13 consecutive generations as flubendiamide selected strain while susceptible strain was field collected strain without exposure to flubediamide for one year. This information has been added in Materials and Methods “2.2 sub-heading insect collection. To make insect collection more clear for laboratory and field conducted experiments, it has been combined under one sub-heading.
- In the table 3, the susceptible, field-pop., unselected and Flubendiamide-sel strain should be clearly described to the reader, because the relationship between these strains are unclear.
Answer: Ok, from table 3, un-selected strain has been deleted to make it clear to the readers. Information regarding field collected; susceptible and selected strain has been added in Materials and Methods Section.
- Cross-resistance refers to the resistance of insects to one particular insecticide that may cause resistance to other insecticides that they have never been exposed to before. So, a selected resistant strain and an unselected reference strain which has same genetic background should be choose for cross-resistance study.
Answer: Yes, we agree with comments. For current studies, cross resistance experiment insects were only exposed to flubendiamide and were having same genetic make up.
- In the part of “2.5 Genetic reciprocal and back crosses”, genetic reciprocal and back crosses, reciprocal crosses and backcross lines should be established by resistance and susceptible strains with the same genetic background. Is your resistant population here screening for the susceptible strain?
Answer: Yes, we agree with comments, and we have added text in revised version (in Materials and Methods: 2.1 Insect collection sub-heading) about developing reference strain without exposure of insecticide (designated as laboratory susceptible strain) from field collected resistant population.
- In the part of “2.7. Degree of dominance Degree of dominance”, how do you calculate the fitness? Please provide detailed methods and references to support the evaluation of the fitness and degree of dominance.
Answer: We have added formula to calculate the degree of dominance (Stone 1968) so that degree of dominance can be calculated. Fitness can be assessed with 0-1 range which is already described under effective dominance formula.
- In the part of 2.8 (line 167-180 of page 4), the references should be cited to support these two formulas.
Answer: Ok, we have added (Sokal and Rohf 1981) and (Lande 1981) as reference of two formulas.
- In the part of “2.9. Statistical Analysis”, Fiducial limit (FL) and Confidence Interval (CI) are two different forms of expression, please keep one form in the manuscript.
Answer: We have added Fiducial Limit (FL) while deleting CI in statistical analysis.
- Revise the title of table 1 to "Resistance levels of Tuta absoluta field populations to flubendiamide".
Answer: We have changed the title of table 1 according to comments.
- In the table 1, revise to "χ²(df)" and merge the two columns of data. When calculate the insecticide resistance ratio of field pest populations, a relatively fixed baseline of sensitivity needs to be established or cited from other study. In your this study, perhaps you can designate a relatively sensitive field population as a sensitive population and establish a sensitive baseline.
Answer: We have added χ²(df) and two columns have been merged of data. We agree with comments of Resistance Ratio (RR). In current studies, we divided the LC50 of Flubendiamide-R field collected strain to LC50 of susceptible strain (collected from non-sprayed field area of tomato where pesticide is not sprayed over years).
- In the table 3, revise LC50 to LC50 (95%FL), revise Slope to Slope ± SE, two RR in the table, maybe you can change one RR to RR'.
Answer: Ok, we have added them in the table 3.
- In the part of “3.3 Maternal effects”, the results of Table 6 have not been presented and explained in the text, so, what can we get from Table 6? Moreover, the backcross information is not completed in the table 6, please revise it.
Answer: We have added the results of table 6 in text and have explained them. Back-cross information has been completed in table 6 and is revised .
- In the table 5, the LC50 values of parental and reciprocal crosses (F1) should be listed here, but the value of resistant strain is absent. Please revise the LC50 to LC50(95%FL), revise Slope to Slope ± SE, and merge the two columns of χ² and df to χ² (df).
Answer: LC50 values in table 5 for resistant strain has been added. Revision for table headings in columns have also been revised and two columns of χ² (df) has been merged.
- In the part of "3.4. Loci influencing inheritance for monogenic fit test", according to the results of Table 7, I can't get any information of statistical analysis.
Answer: P value and df has been added in results section. Reference of Sokal and Rohf (1981) has also been added in Materials and Methods and results section.
- In the part of "3.5. Degree of dominance", I can't understand these findings, does the degree of dominance here is represent the flubendiamide resistance in the Tuta absoluta? If so, how does the degrees of dominance of the resistance can be changed under different concentration of flubendiamide exposure.
Answer: In text and table, we have added the dose dependant response of T. absoluta. Level of complete dominant to incomplete recessive has also been added.
- In the part of "3.6. Cross-resistance", the tables in the manuscript should be numbered in an order. The cross-resistance ratio (CR = LC50 of flubendiamide-resistant strain/LC50 of the unselected strain) has usually been used for evaluating the cross-resistance level between different pesticides.
Answer: We have changed the tables number in revised manuscript. As of comments, we have deleted the field collected strain data and its comparison to susceptible strain for cross resistance. In revised manuscript, LC50 of flubendiamide-resistant strain/LC50 of the unselected strain) has been used for evaluating the cross-resistance level.
Reviewer 2 Report
The manuscript by Zang et al provides interesting results in an important topic. The manuscript presents substantial work that appears well thought through and systematic. Parts of the manuscript are well written and other sections need quite extensive editing to make the findings clear to the reader. In particular, the Discussion appears rushed, has errors and generally unorganised. There also appears to be inconsistencies in the Summary, Abstract and Results, specifically concerning whether resistance is polygenic. The results are very brief with most of the findings presented in the tables. But the tables require more detailed legends and in some cases incomplete.
More specific comments:
3.1 Table 1 - Table title/legend incomplete. Also column headings cut off and cannot be read.
3.2 It is stated that selection is performed for 13 generations. But how were the concentration of Flub chosen. Do the authors really believe that any selection is taking place when % mortality is so low? Therefore it is unclear how higher LC50 are being achieved in Table 3. Is this the reason why the fold resistant was so much lower than Campos et al?
3.3 Cannot read all of Table 5.
3.4 It clearly states that "multiple factors controlling the resistance to flub ..." this disagrees with the Summary and Abstract which states a monofactorial model? But again it appears that text in the table is missing and I do not have enough information to fully interpret the results.
3.6 Why is Table 4 after Tables 5-8? Table legend incomplete.
Discussion: The language usage in the Discussion makes it difficult to follow. Also the organisation is unclear - for example cross-resistance is discussed in Paragraphs 1 and 2, it becomes difficult to know when the authors are discussing their own results or the published literature.
Author Response
The manuscript by Zang et al provides interesting results in an important topic. The manuscript presents substantial work that appears well thought through and systematic. Parts of the manuscript are well written and other sections need quite extensive editing to make the findings clear to the reader. In particular, the Discussion appears rushed, has errors and generally unorganised. There also appears to be inconsistencies in the Summary, Abstract and Results, specifically concerning whether resistance is polygenic. The results are very brief with most of the findings presented in the tables. But the tables require more detailed legends and in some cases incomplete.
Answer: We are thankful for active comments, because of higher actual mortality than expected, mode is polygenic. It has been changed in summary and abstract. Legends have been added in tables. In addition, we have done more revisions on Discussion.
More specific comments:
3.1 Table 1 - Table title/legend incomplete. Also column headings cut off and cannot be read.
Answer: Table 1 title has been changed and one column of df has been merged with x2 column. So that table in revised manuscript is in better format.
3.2 It is stated that selection is performed for 13 generations. But how were the concentration of Flub chosen. Do the authors really believe that any selection is taking place when % mortality is so low? Therefore it is unclear how higher LC50 are being achieved in Table 3. Is this the reason why the fold resistant was so much lower than Campos et al?
Answer: LC50 for G13 was 709.81. Folds resistance (Selected population) in comparison to field collected population was low but higher in comparison to susceptible population.
3.3 Cannot read all of Table 5.
Answer: In revised manuscript, table 5 has been edited. df column has been merged with previous column and selected population data has also been added for better comparison.
3.4 It clearly states that "multiple factors controlling the resistance to flub ..." this disagrees with the Summary and Abstract which states a monofactorial model? But again it appears that text in the table is missing and I do not have enough information to fully interpret the results.
Answer: In revised manuscript, polygenic mode has been added in summary and abstract. Table editing is revised for its better format.
3.6 Why is Table 4 after Tables 5-8? Table legend incomplete.
Answer: We have re-numbered all tables and table legends have also been added.
Discussion: The language usage in the Discussion makes it difficult to follow. Also the organization is unclear - for example cross-resistance is discussed in Paragraphs 1 and 2, it becomes difficult to know when the authors are discussing their own results or the published literature.
Answer: the required changes have been made and the discussion is now more precise and specific to our own results.
Reviewer 3 Report
Authors Zang et al. investigated the genetics of flubendiamide resistance and potential of cross-resistance to other insecticides in the tomato pinworm. Field collected and laboratory selected (Fluben-R) strains showed cross-resistance against chlorantraniliprole, thiamethoxam, permethrin, abamectin and tebufenozide compared to the susceptible strain. Backcross analysis and resistance progression analysis suggest that resistance is controlled by one or more factors, one locus and two alleles maybe involved in resistance. The research design, data analysis and interoperation are scientifical sound. The presentation of results is clear. I only have a few minor comments.
Page 1, line 20, delete “higher occurrence of”
Page 1, line 21, “compared to susceptible” change to “compared to the susceptible strain”
Page 3, line 119, the susceptible strain was also collected from lubendiamide sprayed fields in Faisalabad district? Please clarify it.
Author Response
Authors Zang et al. investigated the genetics of flubendiamide resistance and potential of cross-resistance to other insecticides in the tomato pinworm. Field collected and laboratory selected (Fluben-R) strains showed cross-resistance against chlorantraniliprole, thiamethoxam, permethrin, abamectin and tebufenozide compared to the susceptible strain. Backcross analysis and resistance progression analysis suggest that resistance is controlled by one or more factors, one locus and two alleles maybe involved in resistance. The research design, data analysis and interoperation are scientifical sound. The presentation of results is clear. I only have a few minor comments.
Answer: Thank you for the active comments on our study.
Page 1, line 20, delete “higher occurrence of”
Answer: Done.
Page 1, line 21, “compared to susceptible” change to “compared to the susceptible strain”
Answer: Done.
Page 3, line 119, the susceptible strain was also collected from flubendiamide sprayed fields in Faisalabad district? Please clarify it.
Answer: Susceptible laboratory strain was developed from field collected resistant population, in revised manuscript we have added the detail about strains in materials and methods; insect collection section.
Round 2
Reviewer 1 Report
1. In the part of “2.1. Insect collection”, there are two “susceptible strain” in the paragraph one and two, but they are different. I suggest that you name the resistant and susceptible strain for inheritance and cross resistance studies as “Fluben-sel” and “Unsel”, respectively. In the Table 3, please revise the “Susceptible” and “Flubendiamide-sel” to “Unsel” and “Fluben-sel”, respectively, and delete the data of “Field-pop.”. Please use “Fluben-sel” and “Unsel” in any other please of this manuscript if necessary.
2. In the Table 3, for the “Susceptible (LC50 = 1.29 µg/mL)” and “Field-pop. (LC50 = 47.68 µg/mL)” strain, I just want to know that which one is the “designated susceptible strain was kept on non-sprayed tomato for a year (line 144 of page 3)”? If the “Susceptible (LC50 = 1.29 µg/mL)” is the unselected strain (for a year) from the field collected population, that means the LC50 value of flubendiamide against the field collected population decreased from about 50 µg/mL to 1.29 µg/mL in one yuar? I think this is very important to clarify, because the further cross-resistance and inheritance study all should be based on these two strains (Fluben-sel and Unsel).
3. Revise the title of Table 3 to “The resistance levels of Tuta absoluta to flubendiamide during the selection process”.
4. The purpose of the part of 3.3 to 3.5 is to make clear the inheritance mode of flubendiamide resistance in the Tuta absoluta, that is whether this resistance inherited as autosomes or sex chromosomes, dominance or recessive, monogenic or polygenic. Although some references have been added to the formula in the manuscript, but some points are still hard for me to understand.
For example, in the pare of “2.6. Degree of dominance”,
“Effective dominance (h) was calculated as: h = (W12 -W22) / (W11-W22). The W11, W12, W22 are fitness of homozygous resistant parental strains, heterozygous offspring, homozygous susceptible parental strains respectively. h can vary from 0 (completely recessive); 0.5 (co-dominance), 1 (completely dominant).”
Here, how you calculate the fitness of homozygous resistant parental strains (W11), heterozygous offspring (W12), homozygous susceptible parental strains (W22)?
Moreover, in the pare of “3.5. Degree of dominance”, as you described that “h value shows that resistance was expressed as complete dominant at low concentration, became co-dominant at middle concentration; and at highest concentration it was incomplete recessive (Table 7).” How does the degrees of dominance of the resistance can be changed under different concentration of flubendiamide exposure? Additionally, we can find that “Based on effective dominance h value, resistance was tested for inheritance and patterns of inheritance revealed autosomal and incomplete recessive” in your abstract (line 36-37), so, how do you conclude that it’s incomplete recessive but not complete dominant or co-dominant?
Maybe you can improve your manuscript based on these references below.
a) Wang et al., 2009. Inheritance mode and realized heritability of resistance to imidacloprid in the brown planthopper, Nilaparvata lugens (Stål) (Homoptera: Delphacidae). Pest Management Science, 65: 629–634.
b) He et al., 2009. Genetic analysis of abamectin resistance in Tetranychus cinnabarinus. Pesticide Biochemistry and Physiology, 95: 147–151.
c) Shi et al., 2011. Characterisation of spinosad resistance in the housefly Musca domestica (Diptera: Muscidae). Pest Management Science, 67: 629–634.
5. Cross-resistance refers to the resistance of insects to one particular insecticide that may cause resistance to other insecticides that they have never been exposed to before. According to the results from Table 8, we could not determine whether there was cross-resistance between flubendiamide and other insecticides in the flubendiamide resistance strain of Tuta absoluta.
Please check out this literature below.
Wei et al., 2017. Cross-resistance pattern and basis of resistance in a thiamethoxam-resistant strain of Aphis gossypii Glover. Pesticide Biochemistry and Physiology, 138: 91–96.
Author Response
- In the part of “2.1. Insect collection”, there are two “susceptible strain” in the paragraph one and two, but they are different.I suggest that you name the resistant and susceptible strain for inheritance and cross resistance studies as “Fluben-sel” and “Unsel”, respectively. In the Table 3, please revise the “Susceptible” and “Flubendiamide-sel” to “Unsel” and “Fluben-sel”, respectively, and delete the data of “Field-pop.”. Please use “Fluben-sel” and “Unsel” in any other please of this manuscript if necessary.
Answer: Two susceptible strains were collected from non-sprayed (over several years) fields of tomato, in their vicinity no tomato field existed. Difference existed between both susceptible strains was, this strain was kept in laboratory for a year without exposure to insecticide (Flubendiamide).
In table 3; we have revised the Fluben-sel and Unsel. We agree with your suggestion. Field data from table 3 has been deleted.
- In the Table 3, for the “Susceptible (LC50 = 1.29 µg/mL)” and “Field-pop. (LC50 = 47.68 µg/mL)” strain, I just want to know that which one is the “designated susceptible strain was kept on non-sprayed tomato for a year (line 144 of page 3)”? If the “Susceptible (LC50 = 1.29 µg/mL)” is the unselected strain (for a year) from the field collected population, that means the LC50 value of flubendiamide against the field collected population decreased from about 50 µg/mL to 1.29 µg/mL in one yuar? I think this is very important to clarify, because the further cross-resistance and inheritance study all should be based on these two strains (Fluben-sel and Unsel).
Answer: In revised version, materials and methods section (we have added in brackets) about susceptible strain collection from non-sprayed fields of tomato (no spray on them over years) with no tomato field in the vicinity. These strains were brought and reared in laboratory and kept for one year without flubendiamide exposure.
- Revise the title of Table 3 to “The resistance levels of Tuta absolutato flubendiamide during the selection process”.
Answer: Done.
- The purpose of the part of 3.3 to 3.5 is to make clear the inheritance mode of flubendiamide resistance in the Tuta absoluta, that is whether this resistance inherited as autosomes or sex chromosomes, dominance or recessive, monogenic or polygenic. Although some references have been added to the formula in the manuscript, but some points are still hard for me to understand.
For example, in the pare of “2.6. Degree of dominance”,
“Effective dominance (h) was calculated as: h = (W12 -W22) / (W11-W22). The W11, W12, W22 are fitness of homozygous resistant parental strains, heterozygous offspring, homozygous susceptible parental strains respectively. h can vary from 0 (completely recessive); 0.5 (co-dominance), 1 (completely dominant).”
Here, how you calculate the fitness of homozygous resistant parental strains (W11), heterozygous offspring (W12), homozygous susceptible parental strains (W22)?
Answer: We agree with your comments. In formula, there is Degree of dominance. These sentences have been deleted, because already one formula with reference in text can express the degree of dominance.
Moreover, in the pare of “3.5. Degree of dominance”, as you described that “h value shows that resistance was expressed as complete dominant at low concentration, became co-dominant at middle concentration; and at highest concentration it was incomplete recessive (Table 7).” How does the degrees of dominance of the resistance can be changed under different concentration of flubendiamide exposure?
Answer: We agree with your comments. These sentences have been deleted. Now we have cited references.
Additionally, we can find that “Based on effective dominance h value, resistance was tested for inheritance and patterns of inheritance revealed autosomal and incomplete recessive” in your abstract (line 36-37), so, how do you conclude that it’s incomplete recessive but not complete dominant or co-dominant?
Answer: We agree with your comment, in revised manuscript, these lines have been deleted.
Maybe you can improve your manuscript based on these references below.
- a) Wang et al., 2009. Inheritance mode and realized heritability of resistance to imidacloprid in the brown planthopper, Nilaparvatalugens(Stål) (Homoptera: Delphacidae). Pest Management Science, 65: 629–634.
- b) He et al., 2009. Genetic analysis of abamectin resistance in Tetranychuscinnabarinus. Pesticide Biochemistry and Physiology, 95: 147–151.
- c) Shi et al., 2011. Characterisation of spinosad resistance in the housefly Musca domestica(Diptera: Muscidae). Pest Management Science, 67: 629–634.
Answer: We have cited these three references in revised manuscript.
- Cross-resistance refers to the resistance of insects to one particular insecticide that may cause resistance to other insecticides that they have never been exposed to before.According to the results from Table 8, we could not determine whether there was cross-resistance between flubendiamide and other insecticides in the flubendiamide resistance strain of Tutaabsoluta.
Please check out this literature below.
Wei et al., 2017. Cross-resistance pattern and basis of resistance in a thiamethoxam-resistant strain of Aphis gossypii Glover. Pesticide Biochemistry and Physiology, 138: 91–96.
Answer: We have deleted the word of cross resistance throughout the manuscript; it has been replaced with “concentration mortality response” of Tuta absoluta against different insecticides. This word has also been changed in title; abstract; materials and methods; results; discussion.
We agree with your comments. After reading this research paper, we have changed the word of cross resistance. Because our studies were not having susceptible strain exposed to each insecticide separately.
Reviewer 2 Report
The authors have not answered concerns regarding the lack of selectivity in 3.2 . With such low levels of mortality do they believe that there is any selection?
The organisation is improved but the language usage makes it difficult to interpret.
Author Response
- The authors have not answered concerns regarding the lack of selectivity in 3.2 . With such low levels of mortality do they believe that there is any selection?
Answer: We agree with comments. There are multiple reasons for low mortality.
-When insect shows response to high dose, reason could be frequency of resistance allele increase over time (Helps et al., 2017). In our studies resistance was found controlled by more than one factor with polygenic in nature.
Helps, J. C., Paveley, N. D., & Van Den Bosch, F. (2017). Identifying circumstances under which high insecticide dose increases or decreases resistance selection. Journal of theoretical biology, 428, 153-167.
-In our studies, Tuta absoluta showed higher mortality in G1 and G2 which identifies accuracy of selection process since beginning. Further lowering the mortality/ higher survival rate during selection process can be better understood at molecular level.
-In these studies, dose was chosen from pre-experiments where we got less mortality, so that to get more number of individuals for further selection.
- The organisation is improved but the language usage makes it difficult to interpret.
Answer: We are thankful for comments. Language is improved in revised version. Hopefully this version will be up to the standard.
Round 3
Reviewer 1 Report
1. In the Table 3, “Flubendiamide-unsel” strain has not been described in the part of “Materials and methods”, what is this strain? What’s the difference between “Flubendiamide-unsel” and “designated susceptible strain” (line 136-137)?
2. In the Table 4 and Table 5, the “Susceptible” and “Flubendiamide-sel (G13)” have been used to execute reciprocal cross and obtained two F1 offspring lines; moreover, the backcross lines were obtained from the reciprocal progenies of the F1 cross with the parental strain (RR). The biggest question for me is that the “Flubendiamide-sel (G13)” strain was not established from the “Susceptible” strain by resistance selection. So, we can’t make sure that the genetic backgrounds were similar and homogenous for these two strains, and suitable for reciprocal cross and backcross study. These different insect lines used in this study, including “Fluben-S”, “Flubendiamide-unsel”, “Susceptible”, “Flubendiamide-sel” and “RR”, are still ambiguous for me.
3. The results of Table 5, Table 6, and Table 7 still do not fully support your conclusions of maternal effects, polygenic control and degree of dominance, respectively.
4. Line 272-284, according to you statement and the results, maybe the RR values of Thiamethoxam, Permethrin, Abamectin and Tebufenozide in the Table 8 were calculated by this formula that “LC50 values of Thiamethoxam, Permethrin, Abamectin and Tebufenozide against Flubendiamide-sel (G13) / LC50 value of Flubendiamide against Susceptible”. This is a wrong calculation! The RR values can’t be calculated by two different pesticides.
5. In general, the experimental scheme design of this research is not reasonable enough, and the data analysis can’t well support the corresponding conclusion. Therefore, from the perspective of scientific and rigorous nature, I don’t recommend this article for publication.
Author Response
Comment 1. In the Table 3, “Flubendiamide-unsel” strain has not been described in the part of “Materials and methods”, what is this strain? What’s the difference between “Flubendiamide-unsel” and “designated susceptible strain” (line 136-137)?
Answer: Based on recommendations from previous reviewer (2nd revision) the “Susceptible” was changed to “Fluben-Unsel” in table 3. In revised version, we have changed it to susceptible.
Comment 2. In the Table 4 and Table 5, the “Susceptible” and “Flubendiamide-sel (G13)” have been used to execute reciprocal cross and obtained two F1 offspring lines; moreover, the backcross lines were obtained from the reciprocal progenies of the F1 cross with the parental strain (RR). The biggest question for me is that the “Flubendiamide-sel (G13)” strain was not established from the “Susceptible” strain by resistance selection. So, we can’t make sure that the genetic backgrounds were similar and homogenous for these two strains, and suitable for reciprocal cross and backcross study. These different insect lines used in this study, including “Fluben-S”, “Flubendiamide-unsel”, “Susceptible”, “Flubendiamide-sel” and “RR”, are still ambiguous for me.
Answer: In revised version, two sentences have been added in Materials & Methods section (Genetic reciprocal and back crosses) “Resistant strain was collected from sprayed tomato field, while susceptible strain was maintained in laboratory without insecticide exposure was collected from tomato fields where no spray was used over years. To determine resistance patterns against flubendiamide in Tuta absoluta, susceptible and resistant strains were assumed to be homogenously susceptible and resistant”.
Based on reference Wang et al., 2009 which was recommended by previous reviewer for guidance, susceptible population was collected from different fields located in different cities such as susceptible strain was collected from Hangzhou while resistant strain was collected from Nanjing, while in our studies, both strains were collected from different localities in same district show more homogenous background. Further reviewer said that the flubendiamide-selected strain was not established from susceptible, based on (Shi et al., 2011) and (Wang et al., 2009) recommended by previous reviewer, selected strain was collected from fields of resistant population and selected but not selected from susceptible strain. Exception exists with “SR” strain (Wei et al., 2017) in which selection was done from susceptible strain. So our studies design was similar to Shi et al. 2011 and Wang et al., 2009.
In revised version, we have also edited fluben-unsel to susceptible (table 3).
- a) Wang et al., 2009. Inheritance mode and realized heritability of resistance to imidacloprid in the brown planthopper, Nilaparvatalugens(Stål) (Homoptera: Delphacidae). Pest Management Science, 65: 629–634.
- b) Shi et al., 2011. Characterisation of spinosad resistance in the housefly Musca domestica(Diptera: Muscidae). Pest Management Science, 67: 629–634.
- c) Wei et al., 2017. Cross-resistance pattern and basis of resistance in a thiamethoxam-resistant strain of Aphis gossypii Pesticide Biochemistry and Physiology, 138: 91–96.
Comment 3. The results of Table 5, Table 6, and Table 7 still do not fully support your conclusions of maternal effects, polygenic control and degree of dominance, respectively.
Answer: We agreed with your comments. In revised version, we have edited the conclusion in abstract and summary based on our results.
Comment 4. Line 272-284, according to you statement and the results, maybe the RR values of Thiamethoxam, Permethrin, Abamectin and Tebufenozide in the Table 8 were calculated by this formula that “LC50 values of Thiamethoxam, Permethrin, Abamectin and Tebufenozide against Flubendiamide-sel (G13) / LC50 value of Flubendiamide against Susceptible”. This is a wrong calculation! The RR values can’t be calculated by two different pesticides.
Answer: We agree with comments. Being Resistance Ratio is more important to calculate in cross resistance studies rather than in bioassay toxicity or concentration mortality response studies. So we have deleted Resistance Ratio (RR) column.
Comment 5. In general, the experimental scheme design of this research is not reasonable enough, and the data analysis can’t well support the corresponding conclusion. Therefore, from the perspective of scientific and rigorous nature, I don’t recommend this article for publication.
Answer: We have made efforts to justify our results with conclusions in revised version. Hopefully this revised version will be upto the mark for acceptance in the Journal.